# Plasma Next Generation Sequencing and Droplet Digital-qPCR-Based Quantification of Circulating Cell-Free RNA for Noninvasive Early Detection of Cancer

**DOI:** 10.3390/cancers12020353

**Published:** 2020-02-04

**Authors:** Martin Metzenmacher, Renáta Váraljai, Balazs Hegedüs, Igor Cima, Jan Forster, Alexander Schramm, Björn Scheffler, Peter A. Horn, Christoph A. Klein, Tibor Szarvas, Hennig Reis, Nicola Bielefeld, Alexander Roesch, Clemens Aigner, Volker Kunzmann, Marcel Wiesweg, Jens T. Siveke, Martin Schuler, Smiths S. Lueong

**Affiliations:** 1Department of Medical Oncology, West German Cancer Center, University Hospital Essen, University Duisburg-Essen, Hufelandstrasse 55, 45122 Essen, Germany; Martin.Metzenmacher@uk-essen.de (M.M.); Marcel.Wiesweg@uk-essen.de (M.W.); Martin.Schuler@uk-essen.de (M.S.); 2Division of Thoracic Oncology, University Medicine Essen-Ruhrlandklinik, University Duisburg-Essen, Tüschener Weg 40, 45239 Essen, Germany; 3Department of Dermatology, University Hospital Essen, West German Cancer Center, University Hospital Essen, University Duisburg-Essen, Hufelandstrasse 55, 45122 Essen, Germany; Renata.Varaljai@uk-essen.de (R.V.); Alexander.Roesch@uk-essen.de (A.R.); 4German Cancer Consortium (DKTK), Partner site University Hospital Essen, Hufelandstrasse 55, 45122 Essen, Germany; i.cima@dkfz-heidelberg.de (I.C.); j.forster@dkfz-heidelberg.de (J.F.); Bjoern.Scheffler@uk-essen.de (B.S.); Nicola.Bielefeld@uk-essen.de (N.B.); Jens.Siveke@uk-essen.de (J.T.S.); 5Department of Thoracic Surgery, Ruhrlandklinik, University Duisburg-Essen, D-45239 Essen, Germany; Balazs.Hegedues@rlk.uk-essen.de (B.H.); Clemens.Aigner@rlk.uk-essen.de (C.A.); 6DKFZ-Division Translational Neurooncology at the WTZ, DKTK partner site, University Hospital Essen, 45122 Essen, Germany; 7Chair for Genome Informatics, Department of Human Genetics, University Hospital Essen, University Duisburg-Essen, Hufelandstrasse 55, 45122 Essen, Germany; 8Laboratory for Molecular Oncology, Department of Medical Oncology, West German Cancer Center, University Hospital Essen, University of Duisburg-Essen, 45122 Essen, Germany; alexander.schramm@uni-due.de; 9Institute for Transfusion Medicine, University Hospital Essen, 45122 Essen, Germany; Peter.Horn@uk-essen.de; 10Experimental Medicine and Therapy Research, University of Regensburg, 93053 Regensburg, Germany; Christoph.Klein@klinik.uni-regensburg.de; 11Fraunhofer-Institute for Toxicology and Experimental Medicine, Division of Personalized Tumour Therapy, 93053 Regensburg, Germany; 12Department of Urology, West German Cancer Center, University Hospital Essen, University Duisburg-Essen, 45122 Essen, Germany; Tibor.Szarvas@uk-essen.de; 13Department of Urology, Semmelweis University, H-1085 Budapest, Hungary; 14Institute of Pathology, West German Cancer Center, University Hospital Essen, University Duisburg-Essen, 45147 Essen, Germany; Henning.Reis@uk-essen.de; 15Institute for Developmental Cancer Therapeutics, West German Cancer Center, University Hospital Essen, 45122 Essen, Germany; 16Department of Internal Medicine II, University Hospital Würzburg, 97080 Würzburg, Germany; kunzmann_v@ukw.de

**Keywords:** liquid biopsy, cfRNA, cancer, ddPCR, NGS, POU6F2-AS2, early detection

## Abstract

Early detection of cancer holds high promise for reducing cancer-related mortality. Detection of circulating tumor-specific nucleic acids holds promise, but sensitivity and specificity issues remain with current technology. We studied cell-free RNA (cfRNA) in patients with non-small cell lung cancer (NSCLC; *n* = 56 stage IV, *n* = 39 stages I-III), pancreatic cancer (PDAC, *n* = 20 stage III), malignant melanoma (MM, *n* = 12 stage III-IV), urothelial bladder cancer (UBC, *n* = 22 stage II and IV), and 65 healthy controls by means of next generation sequencing (NGS) and real-time droplet digital PCR (RT-ddPCR). We identified 192 overlapping upregulated transcripts in NSCLC and PDAC by NGS, more than 90% of which were noncoding. Previously reported transcripts (e.g., HOTAIRM1) were identified. Plasma cfRNA transcript levels of POU6F2-AS2 discriminated NSCLC from healthy donors (AUC = 0.82 and 0.76 for stages IV and I–III, respectively) and significantly associated (*p* = 0.017) with the established tumor marker Cyfra 21-1. cfRNA yield and POU6F2-AS transcript abundance discriminated PDAC patients from healthy donors (AUC = 1.0). POU6F2-AS2 transcript was significantly higher in MM (*p* = 0.044). In summary, our findings support further validation of cfRNA detection by RT-ddPCR as a biomarker for early detection of solid cancers.

## 1. Introduction

Lung cancer (LC) is a leading cause of all cancer-related deaths in men and women worldwide, accounting for more than 14% of all newly diagnosed cancers [1,2]. Although the understanding of lung cancer pathobiology has significantly improved over the last decades, disease prognosis continue to remain poor with a five-year survival rate of only 17% and 7% for non-small cell lung cancer (NSCLC) and small cell lung cancers (SCLC), respectively [3,4]. As with other cancers, poor disease prognosis is partially attributed to late stages at diagnosis, given that there are very few early symptoms [5,6]. Indeed, about 67% of all lung cancer cases are diagnosed at a late stage, leading to high mortality rates [6]. Lung cancer treatment is stage-specific. Early stage disease (I-II UICC 8th edition) can be cured by surgical resection, while locally advanced disease requires multimodal treatment, including chemotherapy, radiotherapy, and surgery for selected cases. Recurrent metastatic disease is treated with palliative systemic therapy [7,8,9]. When diagnosed at an early stage, patients with NSCLC have a 5-year survival rate of about 71%, as opposed to less than 2% for patients diagnosed with stage IV disease [10]. Early diagnosis may thus improve patient outcome [11]. 

To this end, low-dose computed tomography (LDCT) was developed for screening in high-risk groups, with a reported sensitivity of about 93% and a 20% reduction in mortality [12,13]. Despite this, its wide application is hampered by consideration of cost effectiveness, high rates of false positives, risk of developing radiation-related cancers, and over-diagnosis [14,15]. Therefore, there is still a considerable unmet need to develop non-invasive complementary biomarkers for early detection of lung cancer.

Currently, protein- and microRNA (miRNA)-based markers are at advanced validation stages or already in clinical use [16]. Carcinogen embryonic antigen (CEA) is one of the most widely used biomarkers for cancer screening in several entities, including lung cancer. Cytokeratin-19 fragment (Cyfra21-1) is also a tumor marker for lung cancer [17,18]. However, high false positive elevated levels of these markers in benign disease and restricted sensitivity limit their use for routine screening for early diagnosis of cancer [6,18,19,20]. Immune-based markers are attractive. However, they show very low sensitivity. For example, one commercially available autoimmune antibodies test showed only 36% sensitivity but 91% specificity in NSCLC [21]. miRNAs have been also tested in several studies on a plethora of platforms, including microarray, next-generation sequencing (NGS), and RT-PCR. While NGS and microarray-based approaches are highly reproducible and specific, they lack the accuracy and sensitivity required for early detection, especially in the context of low copy transcripts [22]. 

The most attractive non-invasive solution for the identification of biomarkers for early detection of lung cancer is through simple blood test [23]. To date, very few studies have addressed the utility of cell-free RNA (cfRNA) for early diagnosis of cancer, despite previous reports highlighting the potential clinical utility of this liquid compartment. Initially identified in patients with malignant melanoma [24] and nasopharyngeal carcinoma [25], cfRNA has been reported in several other cancer entities, including breast cancer [26], colorectal cancer [27,28,29], follicular lymphoma [30], and hepatocellular carcinoma [31]. Recently, cfRNA has been used for non-invasive determination of gestational aged with better performance compared with standard techniques [32]. cfRNA is therefore a potential source of analyte for early detection biomarker discovery. In light of the aforementioned, we herein report on the use of cfRNA for early detection of cancer in solid cancers with an application to lung, pancreas, bladder, and skin cancers. We use a combination of NGS-based cfRNA profiling and real-time digital droplet PCR (RT-ddPCR) for the analysis of cfRNA for early diagnosis of solids cancers.

## 2. Results

### 2.1. Patients and Baseline Characteristics 

In total, we analyzed cfRNA in blood samples from 153 patients with four different cancer entities and 65 healthy donors (Table 1) using total cfRNA-sequencing and RT-ddPCR. First, a pilot cohort comprising 11 stage IV NSCLC patients, 4 stage IV PDAC patients, and 4 healthy donors (cfRNA-seq) was studied. In 12 of 15 sequenced samples, more than 50 million reads were obtained, and about 40 million reads were obtained for three other samples. More than 70% of reads could be uniquely mapped in all samples, and protein-coding genes were covered by about 5% of reads in all sequenced samples (Appendix A). This was followed by a validation cohort of 84 NSCLC, 12 MM, 20 PDAC, 22 UBC patients, and 61 healthy donors. Tissue expression of specific RNA transcripts was studied in formalin-fixed paraffin-embedded (FFPE) tissue samples from 18 NSCLC and 9 healthy lungs by RT-ddPCR.

### 2.2. Epithelial Cells and Monocytes Differentiate Patient and Healthy cfRNA

Most solid tumors, including lung adenocarcinomas, mainly develop from epithelial cells. We hypothesized that, if the tumor contributes substantially to the cfRNA pool in patients, the transcript signal accounted for by epithelial cells in patients should be significantly higher in patients than in controls. 

To test this, we investigated the contribution of different cell types to the cfRNA pool and if contributions from particular cell types could differentiate patient samples from healthy samples. We thus performed a cell-type signal deconvolution in which we included reference maps from 45 different cell types. As seen from the heatmap in Figure 1A, signal contribution was similar from most cell types. However, there was a significantly higher (false discovery rate = 0.04) contribution from epithelial cells in cancer patients compared with healthy controls. Similarly, monocytes contributed significantly (FDR = 0.02) higher in healthy donors compared with patients (Figure 1B,C). Indeed, in a comparison between patient and healthy samples for epithelial, endothelial, and monocytes cell signatures, we identified epithelial cells and monocytes, but not endothelial cells, to show different transcript levels in patients and controls (Figure 1B,C). 

### 2.3. cfRNA Sequencing Identifies Common Transcripts Upregulated in Cancer Patients

In the pilot cfRNA sequencing study, as shown in Appendix A, pairwise differential gene expression between plasma from NSCLC patients and healthy controls led to the identification of 605 upregulated transcripts (log2FC ≥ 1, FDR < 0.05). A total of 390 transcripts were selectively upregulated (log2FC ≥ 1, FDR < 0.05) in cfRNA from PDAC patients (Appendix A). A subset of 192 genes were overlappingly upregulated in plasma cfRNA derived from both PDAC and NSCLC patients (Figure 2a,b). Non-coding transcripts significantly contributed to those commonly upregulated genes, with pseudogenes, antisense transcripts, and long noncoding RNAs representing 49.2%, 19.2%, and 11.9%, respectively. Protein coding genes accounted for only 5.7% (Figure 2c). 

Gene expression data from NSCLC tumors was downloaded from the gene expression omnibus (GEO). Pair-wise differential gene expression was performed to identify genes that were upregulated in tumor tissue compared with non-tumor tissue. We identified 3060 genes to be upregulated in tumor tissue (log2FC ≥ 1 and FDR < 0.05). In order to identify tumor-relevant cfRNA transcripts, we intersected the list of upregulated genes in tumor tissue and plasma. This identified 24 upregulated transcripts (Figure 2d), including the previously reported HOTAIRM1 [33], (Appendix A). HOTAIRM1 was upregulated in plasma but not in tumor tissue from NSCLC patients (Appendix A). Conversely, the NSCLC-associated transcript MALAT1 was highly expressed in plasma from healthy donors but not NSCLC patients (Appendix A).

We identified a previously reported cancer-related long non-coding RNA (lncRNA), POU6F2-AS2, to be upregulated in both lung cancer cfRNA and tumor tissue, as well as other transcripts, such as the lncRNA AC022126.1 (Figure 2d–f). LncRNA POU6F2-AS2 is known to be associated with another solid tumor, esophageal squamous cell carcinoma [34]. Hence, we selected POU6F2-AS2 and AC022126.1 to further explore the utility of detection of cfRNA transcripts as cancer biomarkers.

### 2.4. The lncRNAs POU6F2-AS2 and AC022126.1 Are Upregulated in Plasma from NSCLC Patients

*POU6F2-AS2* was identified among other (Figure 2d) as a cancer-associated transcript by cfRNA-seq (Figure 3A). Using RT-ddPCR, we could confirm *POU6F2-AS2* as a tumor-associated transcript in plasma samples from 45 stage IV NSCLC patients and 39 stage I–III NSCLC patients (Figure 3B). There was no correlation between *POU6F2-AS2* expression and NSCLC stage (Figure 3C,D; AUC = 0.82 and 0.76; for late stage and early stage diseases, respectively) (Appendix A). Furthermore, *POU6F2-AS2* expression was similar in all stages of NSCLC and could not discriminate early stage from late stage disease (AUC = 0.42) (Appendix A). In contrast, the lncRNA AC022126.1 transcripts failed to discriminate cfRNA from cancer patients and healthy donors in our validation cohort (Figure 3E,F).

### 2.5. High POU6F2-AS2 Levels Is Associated with Higher Cyfra 21.1 Levels

Levels of the tumor marker Cyfra21-1 were significantly associated (*p* = 0.017) with higher plasma levels of *POU6F2-AS2* (Table 2). No significant association was found between our selected cfRNA transcripts and tumor stage, CEA, disease histology, smoking status, gender, and age. *POU6F2-AS2* cfRNA transcript levels were not correlated with clinical data for other cancer entities due to small sample numbers.

### 2.6. Serum cfRNA Concentration Correlates with POU6F2-AS2 Expression

Transcript abundance can be affected by the overall RNA yield (expressed in ng/mL) from a sample. Given that equal volumes of different samples may result in varying amounts of total cfRNA, we investigated the effect of input cfRNA amount on absolute transcript counts in plasma and serum. cfRNA yield per unit sample volume was higher in PDAC serum and could discriminate patients from healthy donors (Appendix A). Similarly, in PDAC sera, there was a very strong correlation (*r* = 0.98) between the transcript abundance of *POU6F2-AS* per unit volume and the total cfRNA yield (Appendix A). In NSCLC plasma samples, there was a weak correlation (*r* = 0.33) between total cfRNA yield and *POU6F2-AS2* transcript abundance (Appendix A). Similarly, there was no difference in plasma cfRNA yield between NSCLC at different stages and between patient and healthy controls (Appendix A). 

Furthermore, the total amount of cfRNA isolated per unit volume of sample was about 4-fold lesser in plasma than in serum. This trend was observed in both healthy donor samples and PDAC patient samples (Appendix A). 

### 2.7. The lncRNAs POU6F2-AS2 and AC022126.1 Are Upregulated in NSCLC Tissue

To investigate the tissue expression of candidate transcripts, we used RNA sequencing data from publicly available sources and RT-ddPCR validation on RNA isolated from FFPE NSCLC tumor tissue samples and adjacent normal lung tissue. In publicly available RNA sequencing data from tumor samples, the transcript levels of *POU6F2-AS2* was significantly upregulated in tumor-derived RNA as compared to non-tumor lung tissue (Figure 4A,B). Interestingly, tumor expression of *POU6F2-AS2* was independent of NSCLC stage (Figure 4C). Validation by RT-ddPCR analysis of RNA from FFPE samples revealed higher *POU6F2-AS2* transcript levels in tumor-derived RNA as compared to normal lung RNA (Figure 4D). There was no impact of NSCLC stage in terms of discrimination between tumor and normal tissue RNA (Figure 4E,F).

### 2.8. POU6F2-AS2 Is Upregulated in Additional Cancers

We next asked if *POU6F2-AS2* transcripts are exclusively upregulated in NSCLC plasma. To this end, we analyzed cfRNA isolated from serum samples of 20 additional PDAC, 22 UBC, and 22 healthy serum samples. Additionally, we analyzed cfRNA isolated from 12 MM plasma samples. In cfRNA from PDAC sera, *POU6F2-AS* transcript levels were significantly higher in patients than in healthy controls (Figure 5B). In malignant melanoma, plasma-derived cfRNA, *POU6F2-AS2* transcript levels were significantly higher in patients than controls (Figure 5C). No significant difference was observed in UBC samples and healthy controls (Figure 5D).

## 3. Discussion

Early diagnosis of cancer may improve survival outcome, as it allows for curative surgical intervention. Indeed, in lung cancer, the five-year survival rate of patient diagnosed at early stages is much higher than patients diagnosed at later stages [35,36]. Identification and validation of non-invasive biomarkers for early detection of cancer is, therefore, a considerable unmet need. Ideally, a biomarker should offer high sensitivity and sufficient specificity in a cost- and time-efficient manner to allow for its implementation into routine clinical practice [37]. Early diagnosis of lung cancer using LDCT has been shown to reduce mortality about 20% [38]. However, repeated radiation exposure, high cost, and high false positive rates represent significant challenges, and better alternatives are still needed.

Liquid biopsy-based test have been approved for detection of *EGFR* mutation (^mut^*EGFR*) in metastatic lung cancer [39]. While this represents a significant breakthrough, ^mut^*EGFR* is only present in about 10–40% of all NSCLC cases [40]. Besides, the test targets metastatic patients, thus offering little or no chances for early detection for potential curative intervention. Although cfRNA was first reported to be present in several cancer entities more than a decade ago [24,25,26,27,28,29,30,31], its diagnostic value has not been properly evaluated, especially in cancer. In spite of the fact that several legitimate concerns remain regarding the stability of cfRNA, a recent study highlighted the predictive strengths of cfRNA in pregnancy [32]. This demonstrates that cfRNA sequencing is feasible in the context of biomarker identification. 

The study presented here aimed to explore a previously unexplored liquid biopsy compartment with significant opportunities for biomarker discovery. We used two independent highly sensitive technologies (next generation sequencing and droplet digital PCR) for profiling and absolute quantification of circulating-free and vesicle-encapsulated RNA in serum and plasma samples from four different solid tumor entities. Furthermore, we evaluated the expression of selected candidate transcripts in a large cohort of tumor samples and adjacent non-tumor tissue. We validated the expression pattern of a potential candidate in early and late-stage plasma samples from lung cancer patients, as well as tumor/normal tissue samples. Patient mutation status was not considered. 

Unlike tissue-derived RNA, which is predominantly contributed by cells of a specific organ, cfRNA is usually contributed by several sources as blood circulates through the body. We performed transcript signal deconvolution using reference maps from 45 different cell types and identified a distinctive epithelial source of cancer-associated transcripts. Our study reveals that the majority of transcripts that are upregulated in cancer patient plasma are in fact non-coding transcripts. Indeed, some studies [41,42] have reported non-coding RNAs as liquid biopsy-based biomarkers in cancer. Very few coding transcripts were identified, most probably because of endogenous cellular degradation of these transcripts following translation by the *CNOT* complex [43]. Interestingly, we identified a previously reported lncRNA species (*HOTAIRM1*) that is associated with colorectal cancer [34,44]. Furthermore, we identified a panel of transcripts that are highly expressed in plasma from cancer patients (PDAC and lung cancers). We validated a previously reported cancer-associated transcript *POU6F2-AS2* in plasma and serum samples from lung cancer, pancreatic ductal adenocarcinoma, malignant melanoma, and bladder cancer samples. We showed that plasma and tissue transcript levels of certain transcripts correlate to a high degree, demonstrating specificity of the plasma marker for malignant tissue.

We investigated the performance of a candidate marker in early and late stage lung cancer patients. The transcript levels of *POU6F2-AS2* showed a good diagnostic performance in both early and late stage disease with an AUC of 0.82 and 0.76, respectively. This suggests that cfRNA transcript indeed may serve as a potential lung cancer biomarker. The association of tumor tissue in such discovery studies is of crucial importance, as we found out that the lung cancer associated lncRNA *MALAT1*, which is upregulated in tumor tissue [45] was upregulated in healthy plasma. Since cfRNA arises from diverse cellular sources, differential transcript contribution from different sources into the cfRNA pool may indeed affect individual transcript abundance. Hence, only transcripts with significant or exclusive tumor sources are very likely to be informative. Therefore, we performed cell-type signal deconvolution on our cfRNA sequencing data. We could show that only signals from epithelial origin could distinguish patients from healthy donors. Additionally, there was a significantly high contribution of monocytes to the cfRNA pool in healthy controls compared with patients. Furthermore, plasma total cfRNA abundance was similar in healthy donors and lung cancer patients plasma of all stages. This suggests that cfRNA abundance in plasma does not affect transcript expression patterns. On the other hand, serum derived total cfRNA abundance was significantly higher in PDAC serum samples compared with healthy serum. Although the diagnostic value of such differences was not explored here, it may represent an opportunity to be exploited.

While our study does not aim at evaluating the utility of cfRNA in disease surveillance and treatment monitoring, it does provide first-hand evidence of the usefulness of cfRNA in early diagnosis of solid tumors, especially NSCLC. Most importantly, the cross-entity validation and tumor tissue association of selected candidates strengthens the data and opens new opportunities for biomarker discovery. Finally, the fact that transcriptomic changes occur relatively faster than genomic changes and that cellular transcriptional machinery amplifies analyte molecules improves sensitivity and also allows for detection of patients irrespective of the presence or absence of measurable genetic alterations. 

## 4. Materials and Methods

### 4.1. Study Design and Patient Population

Patients analyzed in this study were prospectively recruited at the different study sites, either within clinical trials or clinical translational studies. Lung cancer patients were recruited at the outpatient unit of the Department of Medical Oncology at the West German Cancer Center, University Hospital Essen (stage IV) and at the Ruhrlandklinik (stage I–III) within the framework of the CEVIR study (https://www.transcanfp7.eu/index.php/abstract/cevir.html). Patients with either borderline or locally advanced but non-metastatic pancreatic ductal adenocarcinoma were recruited within the framework of the NEOPLAP prospective, randomized, open-label, phase II clinical trial study (https://clinicaltrials.gov/ct2/show/NCT02125136). Samples from bladder cancer patients were prospectively collected for an institutional biobank between 2008 and 2013, and samples from melanoma patients were collected within standard biobanking procedures at the Department of Dermatology of the University Hospital in Essen. Samples from healthy blood donors were obtained from blood donors at the Department of Transfusion Medicine, University Hospital Essen.

In our pilot study, we performed total cfRNA plasma sequencing on 11 NSCLC patients, 4 PDAC, and 4 healthy blood donors. In the validation study, we performed RT-ddPCR on cfRNA from 45 stage IV, as well as 39 stage I–III NSCL samples, 20 PDAC, 12 melanoma, 22 bladder cancer, and more than 61 healthy donors (serum/plasma). Additionally, RT-ddPCR was performed on a retrospective collection of 18 (unmatched to plasma) lung tumor FFPE samples and 9 adjacent non-tumor lung tissues. The local institutional review boards approved the studies at all sites, and all participants provided written informed consent for biomedical research allowing for molecular analysis on tissue and plasma samples. (NSCLC: 17-7740-BO, 14-6056-BO, PDAC: 17-7729-BO & 92/14 ff, MM: 16-7132-BO, UBC: 08-3942). All healthy controls included in the study provided consent for the use of their samples for molecular analysis, and ethical approval was obtained from the ethical committee at the Medical Faculty of the University of Essen (17-7729-BO).

### 4.2. Blood Sampling and Plasma/Serum Preparation

Blood samples were collected prior to treatment initiation. For plasma preparation, blood samples were collected in 7.5 mL EDTA blood tubes (ref # 01.1605.001, Sarstedt, Nümbrecht, Germany) and centrifuged at 200× *g* for 10 min at 4 °C. The upper phase was then transferred into 2 mL tubes (cat # 296920064, Neolab, Heidelberg, Germany) and then centrifuged at 800× *g* for 10 min at 4 °C. Finally, the supernatant was aliquoted into 2-mL tubes and then centrifuged at 16,000× *g* for 10 min at 4 °C. The resulting plasma was transferred into new 2-mL tubes and stored at −80 °C until cfRNA extraction. When plasma preparation was not possible immediately, the samples were immediately kept at 4 °C and processed within 2 h of blood draw. Serum was prepared from blood collected in 10-mL BD Vacutainer^®^ plus plastic tubes (cat # 366430, BD, Heidelberg, Germany). Blood samples were allowed to stand at room temperature for 30 min and then centrifuged at 2000× *g* for 10 min at 4 °C. The resulting serum was aliquoted and stored at −80 °C until cfRNA isolation.

### 4.3. cfRNA Isolation and Quantification

Cell-free RNA was isolated using the Plasma/Serum RNA Purification Mini Kit (Cat. 55,000, Norgen Biotek, Canada), which isolates both cfRNA and vesicle-encapsulated RNA, following the manufacturer’s instruction with slight modifications. Briefly, plasma/serum samples were allowed to thaw on ice and then 200 µL of plasma/serum cleared at 16,000 × *g* for 2 min at 4 °C and the supernatant transferred into a new tube. The precleared plasma/serum was then combined with 3 volumes (600 µL) of Lysis buffer A, containing 0.01% β-mercaptoethanol and mixed by vortexing. To one volume of lysate (800 µL), an equal amount of absolute ethanol was added and mixed by vortexing for 5 s. The lysate was then loaded onto a mini column and allowed to stand at room temperature for 10 min for RNA binding. The columns were then centrifuged at 6000 rpm for 3 min. The columns were then washed 3× with 400 µL of wash buffer A and dried by centrifugation at 13,000 rpm for 2 min. RNA was eluted in 25 µL of elution buffer A after incubation on the column for 15 min. The RNA concentration was measured using a Quantus fluorometer (Promega, Madison, Fitchburg WI, USA), and samples were stored at −80 °C until analyzed.

### 4.4. RNA Isolation from Tumor/Non-Tumor Tissue

RNA isolation was performed using the simplyRNA tissue kit (cat# AS1340, Promega Corporation), following manufacturer’s instructions. Briefly, tissue pieces were cut on dry ice with a sterile scalpel blade and placed in a Precellys lysing kit tube (cat# P000912-LYSKO-A, Bertin Corp, Montigny-le Bretonneux, France). Sample homogenization buffer containing 0.02% thioglycerol was added, and the tube was placed into a precooled Precellys device. The Precellys machine was then programmed to rupture hard tissue for 2 min. After the tissue was completely disintegrated, 200 µL of lysis buffer was added to the cell suspension and then transferred into a cartridge of Maxwell SimplyRNA tissue kit (cat# AS1340, Promega Corporation), and 5 µL of DNase 1 was added to the appropriate well of the cartridge. RNA isolation was done following the installed kit-specific protocol and eluted in 60 µL of nuclease-free water. Quantification was carried out using a Quantus fluorometer (Promega), and samples were stored at −80 °C until analyses.

### 4.5. RNA Isolation from FFPE Sections

RNA was isolated from FFPE sections using the Maxwell^®^ RSC RNA FFPE kit (Promega, cat# AS1440), following manufacturer’s instructions. Briefly, 2.0 mm^3^ of tissue sections were placed in a 1.5 mL Eppendorf tube and 300 µL of mineral oil was added and vortexed for 10 s. The samples were then heated to 80 °C for 2 min and brought to room temperature. The samples were digested and lysed and then pellet collected by centrifugation at 10,000× *g* for 20 s. The aqueous phase containing the pellet was heated at 56 °C for 15 min and then at 80 °C for 1 h and then incubated at room temperature for 30 min. Samples were DNA-digested and then loaded into the Maxwell cartridge, and RNA isolation was performed following the RSC RNA FFPE kit protocol in the fully automated system. RNA was eluted in 60 µL of nuclease-free water and stored at −80 °C until use.

### 4.6. In Silico Data Mining

In order to identify tumor-associated transcripts that were upregulated in plasma samples from cancer patients, we downloaded publicly available RNA sequencing data from the gene expression omnibus (GEO). A lung cancer data set GSE81089 comprising of both early and late stage cancers was downloaded [46]. The data set was composed of 199 NSCLC fresh frozen samples and 19 paired non-tumor lung tissues.

### 4.7. Total cfRNA Sequencing

Total RNA sequencing libraries were prepared from 450 pg of total cfRNA using the SMARTer Stranded Total RNA-Seq Kit v2—Pico Input Mammalian (cat# 634412, Takara Bio, Mountain View, CA, USA), after depletion of ribosomal RNA. RNA libraries were sequenced on an Illumina Novaseq 6000 with 100 bp paired end read. Fastq files were quality checked with the Fastqc tool, and the reads were trimmed with Trimmomatic tool. Reads were mapped to the human genome (Grch 38), and features were quantified by means of Htseq counts. Gene expression matrices with raw counts were processed with Deseq2 [47] for differential gene expression and validated with edgeR and LIMMA [48,49].

### 4.8. cfRNA Transcript Deconvolution

Cell-type cfRNA signal deconvolution was performed using reference maps from 45 different cell types, as previously described [50]. A complete list of all cell types included in the deconvolution is presented in Appendix A. We then used random forest ensemble classifier to rank the importance of cell type signals differentiating patient and healthy samples as previously described [51]. Hierarchical clustering and statistical test (Wilcoxon-Mann-Whitney U test) were performed in R environment.

### 4.9. Transcript Quantification by RT-ddPCR 

Two transcripts, *POU6F2-AS2* and *AC022126.1,* both of which had no measurable transcripts in the healthy donors in the cfRNA-seq data, were further analyzed. One of these two transcripts, *POU6F2-AS2* has previously been associated with cancer pathogenesis. Circulating levels of *POU6F2-AS2* and *AC022126.1* transcripts were measured by means of RT-ddPCR using the 1-Step RT-ddPCR Advanced Kit for Probes (Bio-Rad, Hercules, CA, USA) in samples from our validation cohort, which included 45 stage IV lung cancer patients, 39 stage I–III lung cancer patients, 20 PDAC, 22 bladder cancer, and more than 65 healthy control samples. Samples from the validation cohort were both plasma and serum. All reactions were performed in duplicates using 2 µL of cfRNA from each sample. The reaction components were constituted following manufacturers instruction to a final volume of 22 µL, of which 20 µL were used for droplet generation in a QX100™/QX200™ droplet generator (Bio-Rad). RT-ddPCR reactions were performed in a C1000 Touch™ thermocycler (Bio-rad), and droplets were read in a QX100™/QX200™ droplet reader (Bio-Rad). The raw transcript concentration (copies/20 µL reaction) was used to determine the absolute transcript load per ml of plasma/serum using the formula:Copies/mL = C × EV × 1000/ISV × TPV
where 

C = copies per 20 µL reactionEV = elution volumeISV = cfRNA input volumeTPV = total plasma/serum volume used of cfRNA isolation

For tissue-derived samples, RT-ddPCR was performed on 4 ng of total RNA for each sample and data expressed per ng of RNA.

### 4.10. Statistical Analysis 

Students’ *t*-test was used to compare the mean of two groups, and the one-way ANOVA was used to compare three or more groups. Categorical data was analyzed by Fishers’ exact test or by Chi-squared test. Strength of relationship between cfRNA copies and amounts was assessed by Person correlation. Statistical significance was set to a *p* value < 0.05. The diagnostic performance of the marker was evaluated with the ROCR package [52]. Data analysis was performed in the R version 3.6-environment and Graphpad prism version 8.3 (GraphPad Software, Inc, La Jolla, CA, USA). 

## 5. Conclusions

We provided evidence on the utility of cfRNA for early detection of solid tumors irrespective of mutation status and identified candidate transcripts with plasma/tissue mirror images. Our findings support further validation of cfRNA detection by RT-ddPCR as a biomarker for early detection of solid cancers.

## Figures and Tables

**Figure 1 cancers-12-00353-f001:**
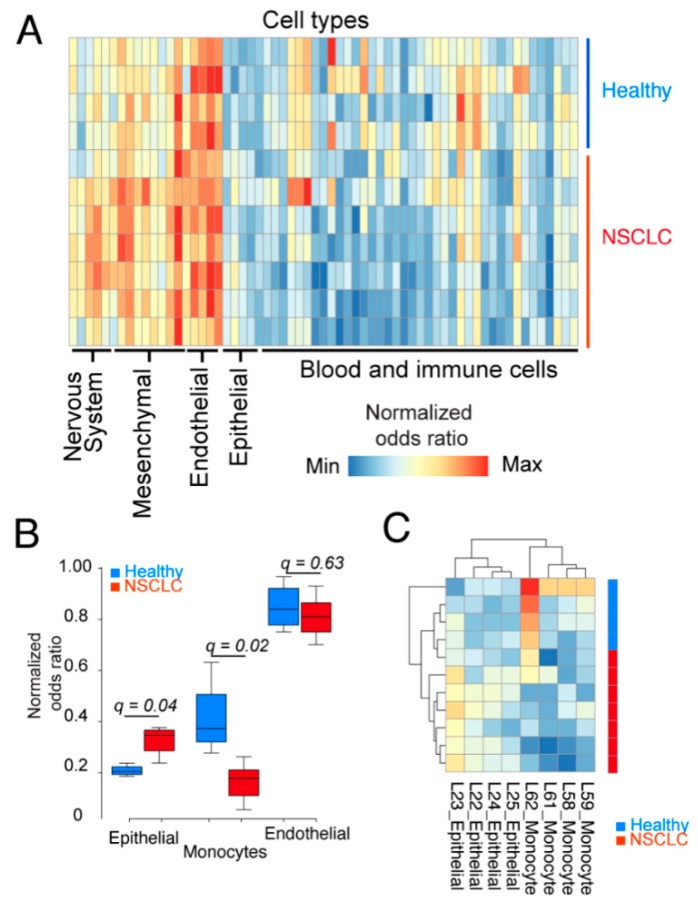
Epithelial cells contribute more to the cfRNA pool of lung cancer patients than controls. (**A**) Cell type deconvolution of 45 different cell types of transcriptomes derived from NSCLC patients (*n* = 7, stage IV) and healthy donors (*n* = 4). Data represents normalized odds ratios comparing number of enriched cell type-specific genes with random enrichment for each sample (rows) and cell type (columns). (**B**) Box plots represent healthy controls (blue) and NSLC patients (red) for epithelial, monocyte, and endothelial cell signatures in plasma. Data are derived from A; q-values represent the false discovery rate-adjusted *p* values. Only epithelial and monocyte signatures had a *q*-value < 0.05. (**C**) Hierarchical clustering of all epithelial and monocyte-specific signature data from (A) separate NSCLC patients.

**Figure 2 cancers-12-00353-f002:**
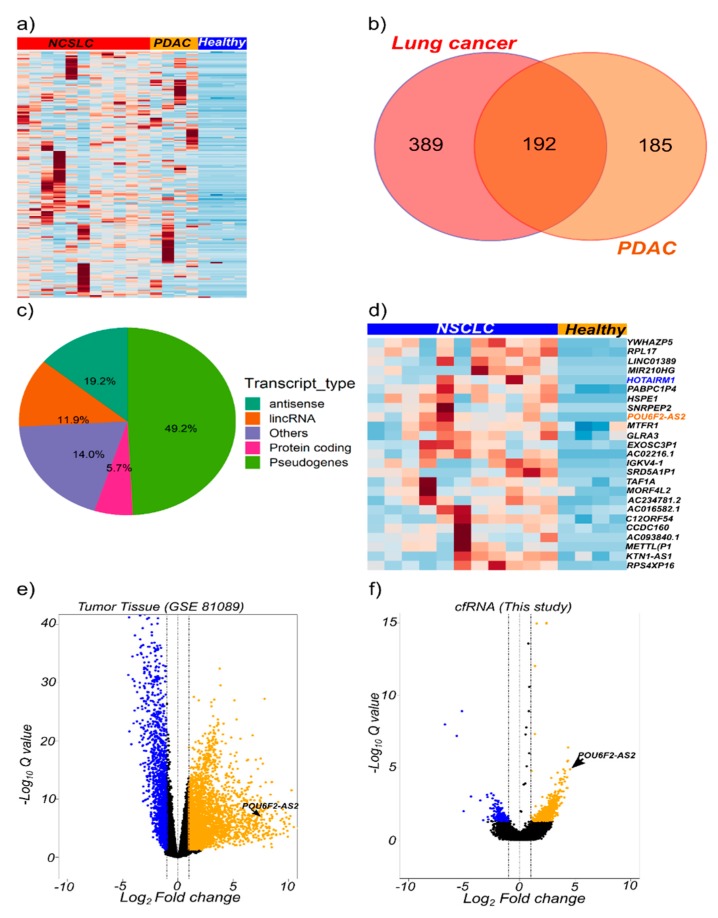
Total cfRNA transcriptome profiling identify potential cancer biomarkers. (**a**) Heatmap of genes upregulated in non-small cell lung cancer (NSCLC; *n* = 11 stage IV patients), and pancreatic ductal adenocarcinoma (PDAC; *n* = 4 stage IV patients) compared with healthy (*n* = 4 blood donors) plasma samples. Heatmap is plotted based on 192 common genes. (**b**) Venn diagram of the number of common upregulated genes in NSCLC and PDAC. (**c**) Pie chart showing the distribution of transcript types across the common upregulated genes. (**d**) Heatmap showing upregulated genes in NSCLC (*n* = 11) and healthy (*n* = 4) plasma samples based on the expression profiles of 24 genes differentially expressed in cfRNA and tumor-derived RNA. Candidate genes were derived from analyses of tumor/healthy tissue (GSE 81089 dataset) and RNA-Seq data from this study. (**e**) Volcano plots for differentially expressed genes in GSE 81089 dataset and (**f**) this study. Genes that passed the log_2_ fold change ≥ 1 and FDR < 0.05 criteria are highlighted in blue (down-regulation) or yellow (up-regulation).

**Figure 3 cancers-12-00353-f003:**
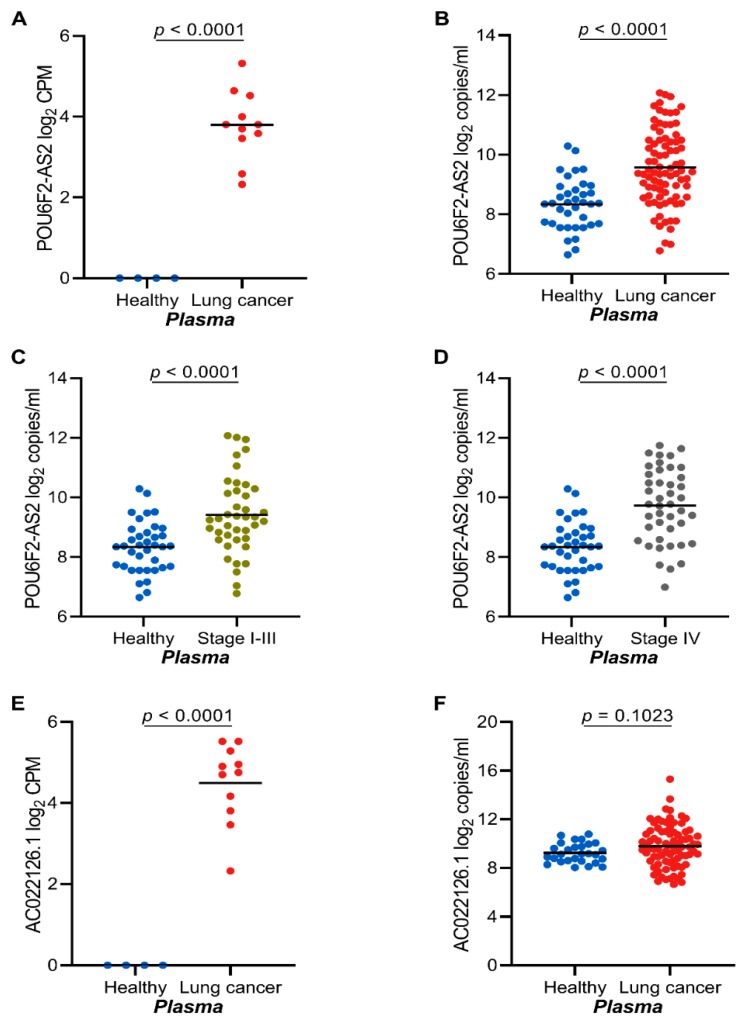
POU6F2-AS2 and AC022126.1 cfRNA transcripts are highly expressed in lung cancer. (**A**) Plasma expression of POU6F2-AS2 as profiled by total cfRNA sequencing in healthy donors (*n* = 4) and NSCLC (*n* = 11). Plasma expression of POU6F2-AS2 as profiled by real-time digital droplet PCR (RT-ddPCR) (**B**). in healthy donors (*n* = 37) and NSCLC (*n* = 84) patients (**C**). in healthy donors (*n* = 37) and early stage NSCLC (*n* = 39) patients (**D**). in healthy donors (*n* = 37) and late stage NSCLC (*n* = 45) patients. Plasma expression of AC022126.1 as profiled by (**E**). total cfRNA sequencing (*n* = 4 heathy donors; *n* = 11 NSCLC patients) and (**F**). by ddPCR (*n* = 28 heathy donors; *n* = 84 NSCLC patients). In (**A**–**F**) scatter dot-plots, the line indicates the mean; Student’s *t*-test *p* values are indicated.

**Figure 4 cancers-12-00353-f004:**
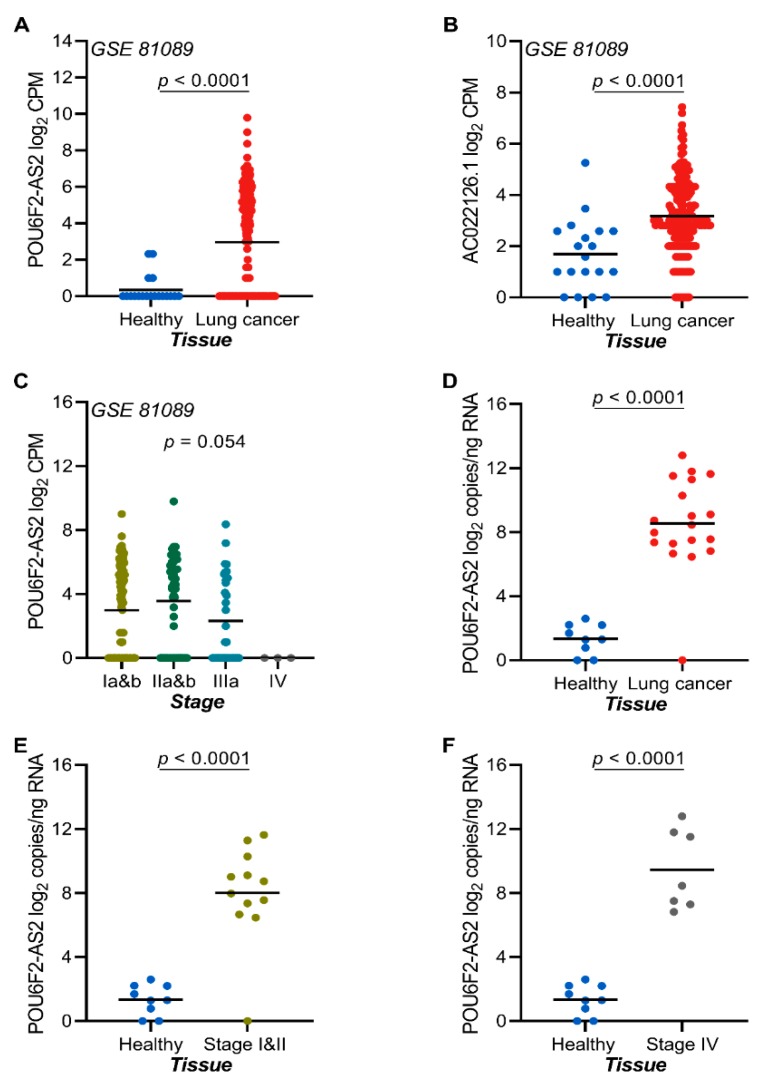
POU6F2-AS2 and AC022126.1 transcripts are highly expressed in lung cancer tissue and is stage independent. (**A**) Tissue expression of POU6F2-AS2 as profiled by RNA sequencing in lung cancer tissue and adjacent non-tumor lung GSE 81089 dataset). (**B**) Tissue expression of AC022126.1 as profiled by RNA sequencing in lung cancer tissue and adjacent non-tumor lung GSE 81089 dataset). (**C**) Association between POU6F2-AS2 expression and tumor stage in NSCLC tumor tissue samples from the GSE 81089 dataset (*n* = 199 tumor tissue samples and 19 non-tumor lung samples). Significance was tested in 1-way ANOVA. (**D**) Tissue expression of POU6F2-AS2 as profiled by RT-ddPCR in healthy donors (*n* = 9) and NSCLC (*n* = 18) patients, (**E**) Tissue expression of POU6F2-AS2 as profiled by RT-ddPCR in healthy donors (*n* = 9) and early stage NSCLC (*n* = 12) patients, and (**F**) Tissue expression of POU6F2-AS2 as profiled by RT-ddPCR in healthy donors (*n* = 9) and late stage NSCLC (*n* = 7) patients. In (**A**–**F**) scatter dot-plots, the line indicates the mean; Student’s *t*-test *p* values are indicated.

**Figure 5 cancers-12-00353-f005:**
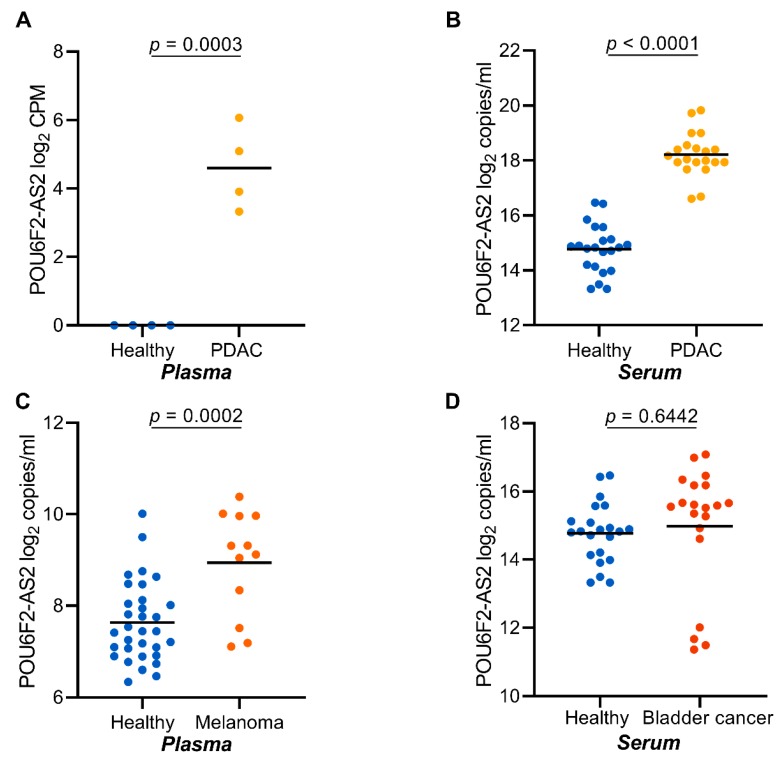
POU6F2-AS2 is highly expressed in cfRNA from other solid tumor entities. (**A**) Plasma expression of POU6F2-AS2 as profiled by total cfRNA sequencing in healthy donors (*n* = 4) and PDAC (*n* = 4). (**B**) Expression of POU6F2-AS2 as profiled by RT-ddPCR in healthy donors’ (*n* = 22) and PDAC (*n* = 20) patients’ sera, (**C**) Expression of POU6F2-AS2 as profiled by RT-ddPCR in healthy donors’ (*n* = 37) and melanoma (*n* =12) patients’ plasma, and (**D**) Expression of POU6F2-AS2 as profiled by RT-ddPCR in healthy donors’ (*n* = 22) and and bladder cancer (*n* = 22) patients’ sera. In (**A**–**D**) scatter dot-plots, the line indicates the mean; Student’s *t*-test *p* values are reported.

**Table 1 cancers-12-00353-t001:** Baseline clinic-pathological parameters of the study population.

Variable	Discovery Cohort		Validation Cohort			
	NSCLC	PDAC	NSCLC	PDAC	Melanoma	Bladder
Total, *n*	11	4	84	20	12	22
Age						
Median (SD)	65 (8.0)	66 (8.7)	66 (9.2)	70 (9.6)	54 (13.5)	70 (10.3)
Gender, *n* (%)						
Male	4 (36)	2 (50)	42 (50)	11 (55)	7 (60)	18 (80)
Female	7 (64)	2 (50)	42 (50)	9 (45)	5 (40)	4 (20)
Smoking, *n* (%)						
Never smoker	-	-	5 (6)	-	-	-
Past smoker	-	-	18 (21)	-	-	-
Current smoker	-	-	14 (17)	-	-	-
Unknown	11 (100)	4 (100)	47 (56)	20 (100)	12 (100)	22 (100)
ECOG status						
0	8 (70)	-	64 (76)	-	10 (83)	-
1	3 (30)	-	16 (19)	-	2 (17)	-
2	-	-	1 (1)	-	-	-
Unknown	-	4 (100)	3 (4)	20 (100)	-	22 (100)
Stage						
Early ^a^	-	1 (25)	39 (46)		-	6 (30)
Late	11 (100)	3 (75)	45 (54)	20 (100)	12 (100)	16 (70)
CEA						
<ULN	1 (1)	-	9 (11)	-	-	-
≥ULN	10 (99)	-	32 (38)	-	-	-
Not done	-	4 (100)	43 (51)	20 (100)	12 (100)	22 (100)
Cyfra21-1						
<ULN	3 (30)	-	5 (6)	-	-	-
≥ULN	8 (70)	-	36 (43)	-	-	-
Not done	-	4 (100)	43 (51)	20 (100)	12 (100)	22 (100)
CA 19.9						
<ULN	-	-	-	5 (25)	-	-
≥ULN	-	2 (50)	-	15 (75)	-	-
Not done	11 (100)	2 (50)	84 (100)	-	12 (100)	22 (100)
LDH						
<ULN	-	-	-	-	10 (83)	-
≥ULN	-	3 (75)	-	-	2 (17)	-
Not done	11 (100)	1 (25)	84 (100)	20 (100)	-	22 (100)

^a^ Stage I-II-III in case of NSCLC, stage I–II in case of PDAC, melanoma and bladder; NSCLC, non-small cell lung cancer; PDAC, pancreatic ductal adenocarcinoma; SD, standard deviation; ECOG, Eastern Cooperative Oncology Group; ULN, upper limit of normal; CEA ULN: 2.6 ng/mL; Cyfra21-1 ULN: 2.1 ng/mL; CA 19.9 ULN: 37 U/mL; LDH ULN: 247 IU/L.

**Table 2 cancers-12-00353-t002:** Association of POU6F2-AS and AC022162.1 expression with baseline characteristics of NSCLC patients in validation cohort.

Variable	*N* (%)	POU6F2-AS2		AC022162.1	
	*n* (%)	Mean	*p*	Mean	*p*
Age					
<66 years	38 (45.2)	9.5	0.686	9.6	0.438
≥66 years	46 (54.8)	9.6		9.9	
Gender					
Male	42 (50)	9.8	0.058	9.9	0.430
Female	42 (50)	9.3		9.6	
Smoking					
Never smoker	5 (6)	9.2	0.833	9.8	0.865
Past smoker	18 (21)	9.5		10.1	
Current smoker	14 (17)	9.6		9.6	
Unknown	47 (56)	9.7		9.7	
ECOG status					
0	64 (76)	9.5	0.517	9.6	0.355
1	16 (19)	9.9		10.3	
2	1 (1)	10.9		11.4	
Unknown	3 (4)	9.7		9.6	
Histology					
Adenocarcinoma	23 (27)	9.6	0.942	9.9	0.913
Non-squamous NSCLC	33 (39)	9.5		9.6	
Squamous NSCLC	21 (25)	9.7		9.9	
Other	7 (8)	9.7		9.7	
Stage					
Early (I-II)	33 (39)	9.4	0.264	9.7	0.854
Late (III-IV)	51 (61)	9.7		9.8	
CEA					
<ULN	9 (11)	9.2	0.371	9.1	0.467
≥ULN	32 (38)	9.8		9.8	
Not done	43 (51)	9.5		9.9	
Cyfra21-1					
<ULN	5 (6)	8.2	0.017	8.3	0.151
≥ULN	36 (43)	9.9		9.9	
Not done	43 (51)	9.5		9.9	

*p*: student’s *t*-test (2-groups) or 1-way ANOVA (3 or more groups); Mean values represent log_2_ RNA copies/ml plasma; CEA ULN (upper limit of normal): 2.6 ng/ML; Cyfra21-1 ULN (upper limit of normal): 2.1 ng/ML; NSCLC, non-small cell lung cancer.

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
