# Peer review of "Plasma Next Generation Sequencing and Droplet Digital-qPCR-Based Quantification of Circulating Cell-Free RNA for Noninvasive Early Detection of Cancer"

_cancers, 2020, doi:10.3390/cancers12020353_

Round 1

Reviewer 1 Report

Minor point

Describe the coverage of genes in each cfRNA sample with RNA sequencing.

Did you Investigate whether there is a difference in expression according to the genetic variation (eg, EGFR, ALK, ROS1 et al.)

page 6, line 133

Log2FC >=FDR < 0.05

Author Response

Describe the coverage of genes in each cfRNA sample with RNA sequencing.

Response: A brief description of the sequencing output and gene coverage has been added to the manuscript as advised (page 3, lines 106-109) and is supported by an additional supplementary figure 1

Did you investigate whether there is a difference in expression according to the genetic variation (eg,EGFR, ALK, ROS1 et al.)

Response: This is definitely an interesting point when it comes to patient stratification for treatment. However, most of our patients did not have information on genetic alterations in tumor tissue. For the few patients with advanced stage cancers, driver mutation analysis was available. However, the number were too small to make any meaningful statistical inference for different groups. A sentence has been added to the discussion section for clarification (page 14 line 276)

page 6, line 133 Log2FC >=FDR < 0.05

Response: We are sorry for this typing mistake. It has been corrected appropriately (page 6, line 140).

Reviewer 2 Report

In this manuscript by Metzenmacher and colleagues, the authors compared cell-free RNA (cfRNA) levels between cancer patients (NSCLC, pancreatic cancer, urothelial cancer, and malignant melanoma) and healthy controls. They used NGS based technique for the Discovery cohort, and then used RT droplet digital qPCR-(ddPCR) based method for the Validation cohort. The authors also evaluated FFPE tissues from 18 NSCLCs and 9 healthy lungs by RT ddPCR.

Comments from the reviewer.

1. The background of healthy donor (as well as the background of 9 healthy lungs, are they from patients with pneumothorax?) should be described. Since cancer patients are usually older, so the reviewer thinks that the authors should show the difference they observed did not come from age difference (or difference in smoking status, etc.) between cancer patients and healthy controls.

2. It is difficult to understand the biological background of the correlation between cfRNA and cell-type signal deconvolution (2.2. Epithelial cells and monocytes differentiate patient and healthy cfRNA). Please add some explanation in this paragraph.  

3. The authors should described more about why the authors chose PU6F2-AS2 and AC022126.1 among genes listed in Figure 2d.

4. What is Non-squamous NSCLC in Table 2? The reviewer thinks it is strange that this subtype is the most common than adenocarcinomas and squamous cell carcinomas.

5. The legend of Figure 3 has some mistakes. Please revise.

6. Why the one patient has no age information (Table 2)??

Author Response

1. The background of healthy donor (as well as the background of 9 healthy lungs, are they from patients with pneumothorax?) should be described. Since cancer patients are usually older, so the reviewer thinks that the authors should show the difference they observed did not come from age difference (or difference in smoking status, etc.) between cancer patients and healthy controls.

Response: We do not have precise age, gender and smoking data from the healthy blood donors. However, the age range of healthy donors in the department of Transfusion medicines from previous studies was from 21-62 years. To address this issue, we have compared the expression of POU6F2-AS2 and HOTAIRM1, both of which are among the 24 cancer associated genes in the patient population. In different age groups, different disease stages and smoking status, there is no significant difference in the expression of both genes in our RT-ddPCR data. A similar analysis has also been performed on the TCGA LUAD data set with more than 500 cases with similar results (supplementary figure 6). Based on these observations, we do not think that such factors would account for the differences observed in the expression of POU6F2-AS2 in cfRNA between healthy donors and patients.

2. It is difficult to understand the biological background of the correlation between cfRNA and cell-type signal deconvolution (2.2. Epithelial cells and monocytes differentiate patient and healthy cfRNA). Please add some explanation in this paragraph.

Response: We have added the paragraph below to the text to highlight the biological rational behind the data presented as suggested (page 4, lines 113-116)

Most solid tumors including lung adenocarcinomas develop from an epithelial cell of origin. We hypothesized, that if the tumor contributes substantially to the cfRNA pool in patients, the transcript signal accounted for by epithelial cells in patients should be significantly higher in patients than in controls.

3. The authors should described more about why the authors chose PU6F2-AS2 and AC022126.1 among genes listed in Figure 2d.

Response: We have added the sentence below to the text to support the use of POU6F2-AS2 and AC022126.1 in the validation study (page 17, lines 415-417)

Two transcripts, POU6F2-AS2 and AC022126.1, both of which had no measurable expression in in the cfRNA-seq analysis of healthy donors, were further analyzed. One of these two transcripts, POU6F2-AS2, has previously been associated with cancer pathogenesis.

4. What is Non-squamous NSCLC in Table 2? The reviewer thinks it is strange that this subtype is the most common than adenocarcinomas and squamous cell carcinomas.

Response:The term “non-squamous NSCLC” is commonly used to differentiate NSCLC with squamous cell histology from the mixed group of other NSCLC histologies (e.g., adenocarcinoma, large cell NSCLC, mixed cellularities, and NSCLC NOS). This is clinically important because all currently approved tyrosine kinase inhibitors address mutations almost exclusively found in “non-squamous NSCLC”. In our cohort characterized by expert thoracic pathology, patients with “non-squamous NSCLC” largely have pulmonary adenocarcinomas.

5. The legend of Figure 3 has some mistakes. Please revise.

Response:We apologize for this discrepancy; the legend has been corrected as suggested

6. Why the one patient has no age information (Table 2)??

Response:This was a patient matching error and has now been corrected in Table 2

Reviewer 3 Report

The authors have performed cfRNA seq in a small subset of patients and then validated their results using a larger group of patients. Despite using a small subset, it is heartening that the authors have been able to identify one transcript POU6F2-AS2 that appears to show uniform expression changes across different data sets-their own and in the public domain. However there are some major concerns that need to be addressed:

- The lack of significant difference in MALAT-1 or HOTAIRM1 between healthy and lung cancer tissue tested in the dataset GSE 81089 are contrary to previous reports. Is this unique to this dataset? Please look at other datasets to confirm these results. In plasma, it is odd that healthy donors have more MALAT-1 than lung cancer patients. This again is contrary to all expectations and the authors have not provided a satisfying explanation. The authors mention that the contribution of monocyte transcripts is higher in healthy donors compared to NSCLC patients. Were the healthy donors free from inflammatory diseases? Since these are most common markers I recommend the authors provide data with the validation subset for these transcripts.

-Lines 142 onwards: the authors remark that they compared upregulated transcripts from NSCLC tumors and plasma and found 24 upregulated transcripts. When the authors compare cfRNA transcripts from two diverse cancer types- PDAC and NSCLC, they find an overwhelming number of 192 common transcripts; yet a comparison between tumor tissue and cfRNA of identical cancer, NSCLC, yields a very small number of common upregulated 24 transcripts. Can the authors provide reasoning? The authors have 11 late stage NSCLC in their validation set. GEO data used seems to comprise of both early and late stage lung cancer. Can authors separate early and late stages in the GEO data and compare their cfRNA data to see if there is improvement in the number of transcripts?

-Lines 207 onwards. The paragraph heading states that POU.._AS2 and AC...1 are upregulated in NSCLC tissue. This title is misleading. The authors observe good concordance with POU6F2-AS2 in their subset of tumor and cfRNA as well as the GEO subsets and some other cancers as well. In contrast AC022126.1 showed upregulated expression in the cfRNA test subset but not in the plasma of validation set, yet was upregulated in the lung cancer GEO dataset. The authors have not provided data for expression of AC022126.1 within their tissue dataset and the conclusion seems to be based on the GEO dataset. Provide data for expression of this transcript in FFPE samples.

-Do the authors used any matched plasma and FFPE samples? It is not clear from the Methods section. It should be stated if they are matched or not. If they are matched, it would be interesting to know if the expressions correlate.

-Of the 24 transcripts that were identified, how many transcripts showed similar mode of expression in the authors dataset, cfRNA and GEO subsets of NSCLSC?

Minor issues:

-The numbers of patients used per cancer type are discrepant, especially UBC and healthy donors- check table 1, lines 107-108 and lines 395-396 and ensure numbers match exactly.

-Methods section does not describe RNA isolation from FFPE samples. Kit used and brief methodology needs to be added.

Author Response

The authors have performed cfRNA seq in a small subset of patients and then validated their results using a larger group of patients. Despite using a small subset, it is heartening that the authors have been able to identify one transcript POU6F2-AS2 that appears to show uniform expression changes across different data sets-their own and in the public domain. However there are some major concerns that need to be addressed:

Response: We appreciate that the reviewer sees the consistency in our findings and his contribution to improving the quality of our study. We have considered all point raised by the reviewer and have provided additional data where possible, corrections where suggested and explanations where necessary.

The lack of significant difference in MALAT-1 or HOTAIRM1 between healthy and lung cancer tissue tested in the dataset GSE 81089 are contrary to previous reports. Is this unique to this dataset? Please look at other datasets to confirm these results. In plasma, it is odd that healthy donors have more MALAT-1 than lung cancer patients. This again is contrary to all expectations and the authors have not provided a satisfying explanation. The authors mention that the contribution of monocyte transcripts is higher in healthy donors compared to NSCLC patients. Were the healthy donors free from inflammatory diseases? Since these are most common markers I recommend the authors provide data with the validation subset for these transcripts.

Response: We have analyzed the expression of MALAT1 and HOTAIRM1 in the LUAD dataset of the TCGA with more than 500 cases and about 60 adjacent normal tissues. This information has been added (supplementary figure 6) as suggested by the reviewer. We observed a significantly higher expression of MALAT1 in patients However, there was still no significant difference in the expression HOTAIRM1 between tumor and non-tumor tissue.  Expression of our candidate transcript POU6F2-AS2 is also significantly different between tumor and non-tumor tissues in this dataset.

Several sources in the body contribute to the cfRNA pool. It is assumed that the contribution of non-hematopoietic cells is less in blood donors without cancer. This, in our opinion, may explain the observation of higher relative monocyte contribution to the cfRNA pool in healthy donors.

With regards to the high expression of MALAT1 in healthy donors, we consider this plausible  since several cell types contribute to the cfRNA pool as opposed to tissue-specific contribution in cancers. MALAT1 in known to be predominantly expressed in human monocytes, and we found monocytes contributing significantly more to the cfRNA pool in healthy donors than in patients. Furthermore, healthy donor samples retained at the Department of Transfusion Medicines are free from known inflammatory conditions.

Lines 142 onwards: the authors remark that they compared upregulated transcripts from NSCLC tumors and plasma and found 24 upregulated transcripts. When the authors compare cfRNA transcripts from two diverse cancer types- PDAC and NSCLC, they find an overwhelming number of 192 common transcripts; yet a comparison between tumor tissue and cfRNA of identical cancer, NSCLC, yields a very small number of common upregulated 24 transcripts. Can the authors provide reasoning? The authors have 11 late stage NSCLC in their validation set. GEO data used seems to comprise of both early and late stage lung cancer. Can authors separate early and late stages in the GEO data and compare their cfRNA data to see if there is improvement in the number of transcripts?

Response: The reviewer’s point is well taken. However, we think that such discrepancies are to be expected for several reasons. 1) Upregulated transcript expression in tumor tissue is contributed by a restricted number of cell types, which in most cases are similar, except for immune cell infiltration and is compared with non-tumor tissue from the same organ. This is not the case with cfRNA, which is contributed by several cell types from several different organs. 2) Different cell types may express the same genes at different levels. This can be seen when expression profiles from bulk tissue and micro-dissected are compared. We therefore consider the similarities between upregulated transcripts in PDAC and NSCLC expected as the main contribution comes from non-cancer cells and thus my represent an entity-independent response.

We have performed the suggested separation and this has improved the concordance, although it is still significantly less than the comparison between PDAC and NSCLC cfRNA. The data has been included in the supplementary table (88 genes between stage I and cfRNA, 92 genes with stage II, 64 genes with stage III and 13 genes with stage IV).

Lines 207 onwards. The paragraph heading states that POU.._AS2 and AC...1 are upregulated in NSCLC tissue. This title is misleading. The authors observe good concordance with POU6F2-AS2 in their subset of tumor and cfRNA as well as the GEO subsets and some other cancers as well. In contrast AC022126.1 showed upregulated expression in the cfRNA test subset but not in the plasma of validation set, yet was upregulated in the lung cancer GEO dataset. The authors have not provided data for expression of AC022126.1 within their tissue dataset and the conclusion seems to be based on the GEO dataset. Provide data for expression of this transcript in FFPE samples.

Response: We agree with the point taken by the reviewer. However, we did not perform transcript quantification for the AC022126.1 transcript in our tumor tissue. The aim of our study was to identify and validate transcripts with concordance in plasma and tumor tissue as potential noninvasive biomarkers. We thus focused our efforts and resources on further validating this transcript in tumor samples, given that it failed to reach statistical significance in the validation cohort. We have revised the heading to better reflect the data presented (page 11, line 215). Performing this analysis is possible, but within the timeframe of this revision is impossible, as the assays are only commercially available and produced on demand with a delivery time of 2-3 weeks.

Do the authors used any matched plasma and FFPE samples? It is not clear from the Methods section. It should be stated if they are matched or not. If they are matched, it would be interesting to know if the expressions correlate.

Response: The FFPE sections were from a retrospective collection and did thus not match to the time of blood withdrawal for our study. This information has now been added to the methods section (page 15 lines 334-336)

Of the 24 transcripts that were identified, how many transcripts showed similar mode of expression in the authors dataset, cfRNA and GEO subsets of NSCLSC?

Response: As shown in figure 2d, all the transcripts indicated were upregulated in the cfRNA sequencing data. These transcripts were selected because they were equally upregulated in the GEO dataset as a whole (which is an RNA sequencing dataset). As the analyses of POU6F2-AS2 and AC022162.1 were validated in a larger sample cohort, we did not analyze the remaining 22 transcripts because of cost limitations.

Minor issues:

The numbers of patients used per cancer type are discrepant, especially UBC and healthy donors- check table 1, lines 107-108 and lines 395-396 and ensure numbers match exactly.

Response: We apologized for these discrepancies with regards to the UBC and healthy donor sample numbers. This has now been corrected to match the text and table (page 3 line 110, page 15 line 334, page 17 lines 421).

Methods section does not describe RNA isolation from FFPE samples. Kit used and brief methodology needs to be added.

Response: The RNA isolation procedure used for FFPE section now is described in the (page 16 lines 380-389)

Round 2

Reviewer 2 Report

The manuscript was well revised. However, the reviewer raises one point about the healthy donor.

The authors responded that "We do not have precise age, gender and smoking data from the healthy blood donors." Did the authors obtain agreement from the healthy donors for their inclusion in this study?

In the manuscript, the authors described that "The local institutional review boards approved the studies at all sites and all participants provided written informed consent for biomedical research allowing for molecular analysis on tissue and plasma samples. (NSCLC: 17-7740-BO, 14-6056-BO, PDAC: 17-7729-BO & 92/14 ff, MM: 16-7132-BO, UBC: 08-3942)". No information about the health donor' written informed consent.

Reviewer 3 Report

The authors have provided compelling answers to the queries.

Regarding figure 4: as mentioned previously the authors have only provided analysis of GEO data for AC022126.1 and not analyzed the expression in their dataset. The authors have mentioned they changed the title heading in line 216 (should be line 217) to reflect this but it does not seem to have been changed. Please remove AC022126.1 from line 217.
